# Fast and Provably Good Seedings for $k$-Means

**Olivier Bachem**
Department of Computer Science
ETH Zurich
olivier.bachem@inf.ethz.ch

**Mario Lucic**
Department of Computer Science
ETH Zurich
lucic@inf.ethz.ch

**S. Hamed Hassani**
Department of Computer Science
ETH Zurich
hamed@inf.ethz.ch

**Andreas Krause**
Department of Computer Science
ETH Zurich
krausea@ethz.ch

## Abstract

*Seeding* – the task of finding initial cluster centers – is critical in obtaining high-quality clusterings for $k$-Means. However, k-means++ seeding, the state of the art algorithm, does not scale well to massive datasets as it is inherently sequential and requires $k$ full passes through the data. It was recently shown that Markov chain Monte Carlo sampling can be used to efficiently approximate the seeding step of k-means++. However, this result requires assumptions on the data generating distribution. We propose a simple yet fast seeding algorithm that produces *provably* good clusterings even *without assumptions* on the data. Our analysis shows that the algorithm allows for a favourable trade-off between solution quality and computational cost, speeding up k-means++ seeding by up to several orders of magnitude. We validate our theoretical results in extensive experiments on a variety of real-world data sets.

## 1 Introduction

k-means++ (Arthur & Vassilvitskii, 2007) is one of the most widely used methods to solve $k$-Means clustering. The algorithm is simple and consists of two steps: In the *seeding* step, initial cluster centers are found using an adaptive sampling scheme called $D^2$-sampling. In the second step, this solution is refined using *Lloyd's algorithm* (Lloyd, 1982), the classic iterative algorithm for $k$-Means.

The key advantages of k-means++ are its strong empirical performance, theoretical guarantees on the solution quality, and ease of use. Arthur & Vassilvitskii (2007) show that k-means++ produces clusterings that are in expectation $\mathcal{O}(\log k)$-competitive with the optimal solution without any assumptions on the data. Furthermore, this theoretical guarantee already holds after the seeding step. The subsequent use of Lloyd's algorithm to refine the solution only guarantees that the solution quality does not deteriorate and that it converges to a *locally* optimal solution in finite time. In contrast, using naive seeding such as selecting data points uniformly at random followed by Lloyd's algorithm can produce solutions that are arbitrarily bad compared to the optimal solution.

The drawback of k-means++ is that it does not scale easily to massive data sets since both its seeding step and every iteration of Lloyd's algorithm require the computation of all pairwise distances between cluster centers and data points. Lloyd's algorithm can be parallelized in the MapReduce framework (Zhao et al., 2009) or even replaced by fast stochastic optimization techniques such as online or mini-batch $k$-Means (Bottou & Bengio, 1994; Sculley, 2010). However, the seeding step requires $k$ inherently sequential passes through the data, making it impractical even for moderate $k$.

This highlights the need for a fast and scalable seeding algorithm. Ideally, it should also retain the theoretical guarantees of k-means++ and provide equally competitive clusterings in practice. Such an approach was presented by Bachem et al. (2016) who propose to approximate k-means++ using a Markov chain Monte Carlo (MCMC) approach and provide a fast seeding algorithm. Under natural assumptions on the data generating distribution, the authors show that the computational complexity of k-means++ can be greatly decreased while retaining the same $\mathcal{O}(\log k)$ guarantee on the solution quality. The drawback of this approach is that these assumptions may not hold and that checking their validity is expensive (see detailed discussion in Section 3).

**Our contributions.** The goal of this paper is to provide fast and competitive seedings for $k$-Means clustering without prior assumptions on the data. As our key contributions, we

(1) propose a simple yet fast seeding algorithm for $k$-Means,
(2) show that it produces provably good clusterings *without assumptions* on the data,
(3) provide stronger theoretical guarantees under assumptions on the data generating distribution,
(4) extend the algorithm to arbitrary distance metrics and various divergence measures,
(5) compare the algorithm to previous results, both theoretically and empirically, and
(6) demonstrate its effectiveness on several real-world data sets.

## 2    Background and related work

We will start by formalizing the problem and reviewing several recent results. Let $\mathcal{X}$ denote a set of $n$ points in $\mathbb{R}^d$. For any finite set $C \subset \mathbb{R}^d$ and $x \in \mathcal{X}$, we define

$$\mathrm{d}(x, C)^2 = \min_{c \in C} \|x - c\|_2^2.$$

The objective of $k$-Means clustering is to find a set $C$ of $k$ cluster centers in $\mathbb{R}^d$ such that the quantization error $\phi_C(\mathcal{X})$ is minimized, where

$$\phi_C(\mathcal{X}) = \sum_{x \in \mathcal{X}} \mathrm{d}(x, C)^2.$$

We denote the optimal quantization error with $k$ centers by $\phi_{OPT}^k(\mathcal{X})$, the mean of $\mathcal{X}$ by $\mu(\mathcal{X})$, and the variance of $\mathcal{X}$ by $\mathrm{Var}(\mathcal{X}) = \sum_{x \in \mathcal{X}} \mathrm{d}(x, \mu(\mathcal{X}))^2$. We note that $\phi_{OPT}^1(\mathcal{X}) = \mathrm{Var}(\mathcal{X})$.

**$D^2$-sampling.** Given a set of centers $C$, the $D^2$-sampling strategy, as the name suggests, is to sample each point $x \in \mathcal{X}$ with probability proportional to the squared distance to the selected centers,

$$p(x \mid C) = \frac{\mathrm{d}(x, C)^2}{\sum_{x' \in X} \mathrm{d}(x', C)^2}. \tag{1}$$

The seeding step of k-means++ builds upon $D^2$-sampling: It first samples an initial center uniformly at random. Then, $k - 1$ additional centers are sequentially added to the previously sampled centers using $D^2$-sampling. The resulting computational complexity is $\Theta(nkd)$, as for each $x \in \mathcal{X}$ the distance $\mathrm{d}(x, C)^2$ in (1) needs to be updated whenever a center is added to $C$.

**Metropolis-Hastings.** The Metropolis-Hastings algorithm (Hastings, 1970) is a MCMC method for sampling from a probability distribution $p(x)$ whose density is known only up to constants. Consider the following variant that uses an independent proposal distribution $q(x)$ to build a Markov chain: Start with an arbitrary initial state $x_1$ and in each iteration $j \in [2, \ldots, m]$ sample a candidate $y_j$ using $q(x)$. Then, either accept this candidate (i.e., $x_j = y_j$) with probability

$$\pi(x_{j-1}, y_j) = \min\left(\frac{p(y_j)}{p(x_{j-1})} \frac{q(x_{j-1})}{q(y_j)}, 1\right) \tag{2}$$

or reject it otherwise (i.e., $x_j = x_{j-1}$). The stationary distribution of this Markov chain is $p(x)$. Hence, for $m$ sufficiently large, the distribution of $x_m$ is approximately $p(x)$.

**Approximation using MCMC (K-MC$^2$).** Bachem et al. (2016) propose to speed up k-means++ by replacing the *exact $D^2$-sampling* in (1) with a fast approximation based on MCMC sampling. In each iteration $j \in [2, 3, \ldots, k]$, one constructs a Markov chain of length $m$ using the Metropolis-Hasting

algorithm with an independent and uniform proposal distribution $q(x) = 1/n$. The key advantage is that the acceptance probability in (2) only depends on $\mathrm{d}(y_j, C)^2$ and $\mathrm{d}(x_{j-1}, C)^2$ since

$$\min \left( \frac{p(y_j)}{p(x_{j-1})} \frac{q(x_{j-1})}{q(y_j)}, 1 \right) = \min \left( \frac{\mathrm{d}(y_j, C)^2}{\mathrm{d}(x_{j-1}, C)^2}, 1 \right).$$

Critically, in each of the $k-1$ iterations, the algorithm *does not require* a full pass through the data, but only needs to compute the distances between $m$ points and up to $k-1$ centers. As a consequence, the complexity of K-MC$^2$ is $\mathcal{O}(mk^2d)$ compared to $\mathcal{O}(nkd)$ for k-means++ seeding.

To bound the quality of the solutions produced by K-MC$^2$, Bachem et al. (2016) analyze the mixing time of the described Markov chains. To this end, the authors define the two data-dependent quantities:

$$\alpha(\mathcal{X}) = \max_{x \in \mathcal{X}} \frac{\mathrm{d}(x, \mu(\mathcal{X}))^2}{\sum_{x' \in \mathcal{X}} \mathrm{d}(x', \mu(\mathcal{X}))^2}, \quad \text{and} \quad \beta(\mathcal{X}) = \frac{\phi_{OPT}^1(\mathcal{X})}{\phi_{OPT}^k(\mathcal{X})}. \tag{3}$$

In order to bound each term, the authors assume that the data is generated i.i.d. from a distribution $F$ and impose two conditions on $F$. First, they assume that $F$ exhibits exponential tails and prove that in this case $\alpha(\mathcal{X}) \in \mathcal{O}(\log^2 n)$ with high probability. Second, they assume that "$F$ is approximately uniform on a hypersphere". This in turn implies that $\beta(\mathcal{X}) \in \mathcal{O}(k)$ with high probability. Under these assumptions, the authors prove that the solution generated by K-MC$^2$ is in expectation $\mathcal{O}(\log k)$-competitive with the optimal solution if $m \in \Theta(k \log^2 n \log k)$. In this case, the total computational complexity of K-MC$^2$ is $\mathcal{O}(k^3 d \log^2 n \log k)$ which is *sublinear* in the number of data points.

**Other related work.** A survey on seeding methods for $k$-Means was provided by Celebi et al. (2013). $D^2$-sampling and k-means++ have been extensively studied in the literature. Previous work was primarily focused on related algorithms (Arthur & Vassilvitskii, 2007; Ostrovsky et al., 2006; Jaiswal et al., 2014, 2015), its theoretical properties (Ailon et al., 2009; Aggarwal et al., 2009) and bad instances (Arthur & Vassilvitskii, 2007; Brunsch & Röglin, 2011). As such, these results are complementary to the ones presented in this paper.

An alternative approach to scalable seeding was investigated by Bahmani et al. (2012). The authors propose the k-means$\|$ algorithm that retains the same $\mathcal{O}(\log k)$ guarantee in expectation as k-means++. k-means$\|$ reduces the number of sequential passes through the data to $\mathcal{O}(\log n)$ by oversampling cluster centers in each of the rounds. While this allows one to parallelize each of the $\mathcal{O}(\log n)$ rounds, it also increases the total computational complexity from $\mathcal{O}(nkd)$ to $\mathcal{O}(nkd \log n)$. This method is feasible if substantial computational resources are available in the form of a cluster. Our approach, on the other hand, has an orthogonal use case: It aims to efficiently approximate k-means++ seeding with a substantially lower complexity.

## 3 Assumption-free K-MC$^2$

Building on the MCMC strategy introduced by Bachem et al. (2016), we propose an algorithm which addresses the drawbacks of the K-MC$^2$ algorithm, namely:

(1) The theoretical results of K-MC$^2$ hold only if the data is drawn independently from a distribution satisfying the assumptions stated in Section 2. For example, the results do not extend to heavy-tailed distributions which are often observed in real world data.

(2) Verifying the assumptions, which in turn imply the required chain length, is computationally hard and potentially more expensive than running the algorithm. In fact, calculating $\alpha(\mathcal{X})$ already requires two full passes through the data, while computing $\beta(\mathcal{X})$ is NP-hard.

(3) Theorem 2 of Bachem et al. (2016) does not characterize the tradeoff between $m$ and the expected solution quality: It is only valid for the specific choice of chain length $m = \Theta(k \log^2 n \log k)$. As a consequence, if the assumptions do not hold, we obtain no theoretical guarantee with regards to the solution quality. Furthermore, the constants in Theorem 2 are not known and may be large.

Our approach addresses these shortcomings using three key elements. Firstly, we provide a proposal distribution that renders the assumption on $\alpha(\mathcal{X})$ obsolete. Secondly, a novel theoretic analysis allows us to obtain theoretical guarantees on the solution quality even without assumptions on $\beta(\mathcal{X})$. Finally, our results characterize the tradeoff between increasing the chain length $m$ and improving the expected solution quality.

---
**Algorithm 1** ASSUMPTION-FREE K-MC$^2$ (AFK-MC$^2$)
---
**Require:** Data set $\mathcal{X}$, # of centers $k$, chain length $m$
    *// Preprocessing step*
 1:  $c_1 \leftarrow$ Point uniformly sampled from $\mathcal{X}$
 2:  **for all** $x \in \mathcal{X}$ **do**
 3:     $q(x) \leftarrow \frac{1}{2} \, \mathrm{d}(x, c_1)^2 / \sum_{x' \in \mathcal{X}} \mathrm{d}(x', c_1)^2 + \frac{1}{2n}$
    *// Main loop*
 4:  $C_1 \leftarrow \{c_1\}$
 5:  **for** $i = 2, 3, \ldots, k$ **do**
 6:     $x \leftarrow$ Point sampled from $\mathcal{X}$ using $q(x)$
 7:     $d_x \leftarrow \mathrm{d}(x, C_{i-1})^2$
 8:     **for** $j = 2, 3, \ldots, m$ **do**
 9:       $y \leftarrow$ Point sampled from $\mathcal{X}$ using $q(y)$
10:       $d_y \leftarrow \mathrm{d}(y, C_{i-1})^2$
11:       **if** $\frac{d_y q(x)}{d_x q(y)} > \mathrm{Unif}(0, 1)$ **then** $x \leftarrow y, d_x \leftarrow d_y$
12:     $C_i \leftarrow C_{i-1} \cup \{x\}$
13:  **return** $C_k$
---

**Proposal distribution.** We argue that the choice of the proposal distribution is critical. Intuitively, the uniform distribution can be a very bad choice if, in any iteration, the true $D^2$-sampling distribution is "highly" nonuniform. We suggest the following proposal distribution: We first sample a center $c_1 \in \mathcal{X}$ uniformly at random and define for all $x \in \mathcal{X}$ the nonuniform proposal

$$q(x \mid c_1) = \frac{1}{2} \underbrace{\frac{\mathrm{d}(x, c_1)^2}{\sum_{x' \in \mathcal{X}} \mathrm{d}(x', c_1)^2}}_{(A)} + \frac{1}{2} \underbrace{\frac{1}{|\mathcal{X}|}}_{(B)} . \tag{4}$$

The term (A) is the true $D^2$-sampling distribution with regards to the first center $c_1$. For any data set, it ensures that we start with the best possible proposal distribution in the second iteration. We will show that this proposal is sufficient even for later iterations, rendering any assumptions on $\alpha$ obsolete. The term (B) regularizes the proposal distribution and ensures that the mixing time of K-MC$^2$ is always matched up to a factor of two.

**Algorithm.** Algorithm 1 details the proposed fast seeding algorithm ASSUMPTION-FREE K-MC$^2$. In the preprocessing step, it first samples an initial center $c_1$ uniformly at random and then computes the proposal distribution $q(\cdot \mid c_1)$. In the main loop, it then uses independent Markov chains of length $m$ to sample centers in each of the $k - 1$ iterations. The complexity of the main loop is $\mathcal{O}(mk^2 d)$.

The preprocessing step of ASSUMPTION-FREE K-MC$^2$ requires a single pass through the data to compute the proposal $q(\cdot \mid c_1)$. There are several reasons why this additional complexity of $\mathcal{O}(nd)$ is not an issue in practice: (1) The preprocessing step only requires a single pass through the data compared to $k$ passes for the seeding of k-means++. (2) It is easily parallelized. (3) Given random access to the data, the proposal distribution can be calculated online when saving or copying the data. (4) As we will see in Section 4, the effort spent in the preprocessing step pays off: It often allows for shorter Markov chains in the main loop. (5) Computing $\alpha(\mathcal{X})$ to verify the first assumption of K-MC$^2$ is already more expensive than the preprocessing step of ASSUMPTION-FREE K-MC$^2$.

**Theorem 1.** *Let $\epsilon \in (0, 1)$ and $k \in \mathbb{N}$. Let $\mathcal{X}$ be any set of $n$ points in $\mathbb{R}^d$ and $C$ be the output of Algorithm 1 with $m = 1 + \frac{8}{\epsilon} \log \frac{4k}{\epsilon}$. Then, it holds that*

$$\mathbb{E}\left[\phi_C(\mathcal{X})\right] \leq 8(\log_2 k + 2)\phi_{OPT}^k(\mathcal{X}) + \epsilon \, \mathrm{Var}(\mathcal{X}).$$

*The computational complexity of the preprocessing step is $\mathcal{O}(nd)$ and the computational complexity of the main loop is $\mathcal{O}\left(\frac{1}{\epsilon} k^2 d \log \frac{k}{\epsilon}\right)$.*

This result shows that ASSUMPTION-FREE K-MC$^2$ produces provably good clusterings for arbitrary data sets without assumptions. The guarantee consists of two terms: The first term, i.e., $8(\log_2 k + 2)\phi_{OPT}^k(\mathcal{X})$, is the theoretical guarantee of k-means++. The second term, $\epsilon \, \mathrm{Var}(\mathcal{X})$, quantifies the potential additional error due to the approximation. The variance is a natural notion as the mean is the optimal quantizer for $k = 1$. Intuitively, the second term may be interpreted as a *scale-invariant and additive approximation error*.

Theorem 1 directly characterizes the tradeoff between improving the solution quality and the resulting increase in computational complexity. As $m$ is increased, the solution quality converges to the theoretical guarantee of k-means++. At the same time, even for smaller chain lengths $m$, we obtain a provable bound on the solution quality. In contrast, the guarantee of K-MC$^2$ on the solution quality only holds for a specific choice of $m$.

For completeness, ASSUMPTION-FREE K-MC$^2$ may also be analyzed under the assumptions made in Bachem et al. (2016). While for K-MC$^2$ the required chain length $m$ is linear in $\alpha(\mathcal{X})$, ASSUMPTION-FREE K-MC$^2$ does not require this assumption. In fact, we will see in Section 4 that this lack of dependence of $\alpha(\mathcal{X})$ leads to a better empirical performance. If we assume $\beta(\mathcal{X}) \in \mathcal{O}(k)$, we obtain the following result similar to the one of K-MC$^2$ (albeit with a shorter chain length $m$).

**Corollary 1.** *Let $k \in \mathbb{N}$ and $\mathcal{X}$ be a set of $n$ points in $\mathbb{R}^d$ satisfying $\beta(\mathcal{X}) \in \mathcal{O}(k)$. Let $C$ be the output of Algorithm 1 with $m = \Theta(k \log k)$. Then it holds that*

$$\mathbb{E}\left[\phi_C(\mathcal{X})\right] \leq 8(\log_2 k + 3)\phi_{OPT}^k(\mathcal{X}).$$

*The computational complexity of the preprocessing is $\mathcal{O}(nd)$ and the computational complexity of the main loop is $\mathcal{O}(k^3 d \log k)$.*

## 3.1 Proof sketch for Theorem 1

In this subsection, we provide a sketch of the proof of Theorem 1 and defer the full proof to Section A of the supplementary materials. Intuitively, we first bound how well a single Markov chain approximates one iteration of exact $D^2$-sampling. Then, we analyze how the approximation error accumulates across iterations and provide a bound on the expected solution quality.

For the first step, consider any set $C \subseteq \mathcal{X}$ of previously sampled centers. Let $c_1 \in C$ denote the first sampled center that was used to construct the proposal distribution $q(x \mid c_1)$ in (4). In a single iteration, we would ideally sample a new center $x \in \mathcal{X}$ using $D^2$-sampling, i.e., from $p(x \mid C)$ as defined in (1). Instead, Algorithm 1 constructs a Markov chain to sample a new center $x \in \mathcal{X}$ as the next cluster center. We denote by $\tilde{p}_m^{c_1}(x \mid C)$ the implied probability of sampling a point $x \in \mathcal{X}$ using this Markov chain of length $m$.

The following result shows that in any iteration either $C$ is $\epsilon_1$-competitive compared to $c_1$ or the Markov chain approximates $D^2$-sampling well in terms of total variation distance[1].

**Lemma 1.** *Let $\epsilon_1, \epsilon_2 \in (0, 1)$ and $c_1 \in \mathcal{X}$. Consider any set $C \subseteq \mathcal{X}$ with $c_1 \in C$. For $m \geq 1 + \frac{2}{\epsilon_1} \log \frac{1}{\epsilon_2}$, at least one of the following holds:*

*(i)*    $\phi_C(\mathcal{X}) < \epsilon_1 \phi_{c_1}(\mathcal{X})$, *or*

*(ii)*    $\|p(\cdot \mid C) - \tilde{p}_m^{c_1}(\cdot \mid C)\|_{\text{TV}} \leq \epsilon_2$.

In the second step, we bound the expected solution quality of Algorithm 1 based on Lemma 1. While the full proof requires careful propagation of errors across iterations and a corresponding inductive argument, the intuition is based on distinguishing between two possible cases of sampled solutions.

First, consider the realizations of the solution $C$ that are $\epsilon_1$-competitive compared to $c_1$. By definition, $\phi_C(\mathcal{X}) < \epsilon_1 \phi_{c_1}(\mathcal{X})$. Furthermore, the expected solution quality of these realizations can be bounded by $2\epsilon_1 \text{Var}(\mathcal{X})$ since $c_1$ is chosen uniformly at random and hence in expectation $\phi_{c_1}(\mathcal{X}) \leq 2 \text{Var}(\mathcal{X})$.

Second, consider the realizations that are not $\epsilon_1$-competitive compared to $c_1$. Since the quantization error is non-increasing in sampled centers, Lemma 1 implies that all $k - 1$ Markov chains result in a good approximation of the corresponding $D^2$-sampling. Intuitively, this implies that the approximation error in terms of total variation distance across all $k-1$ iterations is at most $\epsilon_2(k-1)$. Informally, the expected solution quality is thus bounded with probability $1 - \epsilon_2(k-1)$ by the expected quality of k-means++ and with probability $\epsilon_2(k-1)$ by $\phi_{c_1}(\mathcal{X})$.

Theorem 1 can then be proven by setting $\epsilon_1 = \epsilon/4$ and $\epsilon_2 = \epsilon/4k$ and choosing $m$ sufficiently large.

Table 1: Data sets used in experimental evaluation

| DATA SET | N | D | K | EVAL | $\alpha(\mathcal{X})$ |
|---|---|---|---|---|---|
| CSN (EARTHQUAKES) | 80,000 | 17 | 200 | T | 546 |
| KDD (PROTEIN HOMOLOGY) | 145,751 | 74 | 200 | T | 1,268 |
| RNA (RNA SEQUENCES) | 488,565 | 8 | 200 | T | 69 |
| SONG (MUSIC SONGS) | 515,345 | 90 | 2,000 | H | 526 |
| SUSY (SUPERSYM. PARTICLES) | 5,000,000 | 18 | 2,000 | H | 201 |
| WEB (WEB USERS) | 45,811,883 | 5 | 2,000 | H | 2 |

Table 2: Relative error of ASSUMPTION-FREE K-MC$^2$ and K-MC$^2$ in relation to `k-means++`.

| | CSN | KDD | RNA | SONG | SUSY | WEB |
|---|---|---|---|---|---|---|
| K-MEANS++ | 0.00% | 0.00% | 0.00% | 0.00% | 0.00% | 0.00% |
| RANDOM | 399.54% | 314.78% | 915.46% | 9.67% | 4.30% | 107.57% |
| K-MC$^2$ ($m=20$) | 65.34% | 31.91% | 32.51% | 0.41% | -0.03% | 0.86% |
| K-MC$^2$ ($m=100$) | 14.81% | 3.39% | 9.84% | 0.04% | -0.08% | -0.01% |
| K-MC$^2$ ($m=200$) | 5.97% | 0.65% | 5.48% | 0.02% | -0.04% | 0.09% |
| AFK-MC$^2$ ($m=20$) | 1.45% | -0.12% | 8.31% | 0.01% | 0.00% | 1.32% |
| AFK-MC$^2$ ($m=100$) | 0.25% | -0.11% | 0.81% | -0.02% | -0.06% | 0.06% |
| AFK-MC$^2$ ($m=200$) | 0.24% | -0.03% | -0.29% | 0.04% | -0.05% | -0.16% |

## 3.2 Extension to other clustering problems

While we only consider $k$-Means clustering and the Euclidean distance in this paper, the results are more general. They can be directly applied, by transforming the data, to any metric space for which there exists a global isometry on Euclidean spaces. Examples would be the Mahalanobis distance and Generalized Symmetrized Bregman divergences (Acharyya et al., 2013).

The results also apply to arbitrary distance measures (albeit with different constants) as $D^2$-sampling can be generalized to arbitrary distance measures (Arthur & Vassilvitskii, 2007). However, $\mathrm{Var}(\mathcal{X})$ needs to be replaced by $\phi^1_{OPT}(\mathcal{X})$ in Theorem 1 since the mean may not be the optimal quantizer (for $k = 1$) for a different distance metric. The proposed algorithm can further be extended to different potential functions of the form $\| \cdot \|^l$ and used to approximate the corresponding $D^l$-sampling (Arthur & Vassilvitskii, 2007), again with different constants. Similarly, the results also apply to `bregman++` (Ackermann & Blömer, 2010) which provides provably competitive solutions for clustering with a broad class of Bregman divergences (including the KL-divergence and Itakura-Saito distance).

## 4 Experimental results

In this section[2], we empirically validate our theoretical results and compare the proposed algorithm ASSUMPTION-FREE K-MC$^2$ (AFK-MC$^2$) to three alternative seeding strategies: (1) RANDOM, a "naive" baseline that samples $k$ centers from $\mathcal{X}$ uniformly at random, (2) the full seeding step of `k-means++`, and (3) K-MC$^2$. For both ASSUMPTION-FREE K-MC$^2$ and K-MC$^2$, we consider the different chain lengths $m \in \{1, 2, 5, 10, 20, 50, 100, 150, 200\}$.

Table 1 shows the six data sets used in the experiments with their corresponding values for $k$. We choose an experimental setup similar to Bachem et al. (2016): For half of the data sets, we both train the algorithm and evaluate the corresponding solution on the full data set (denoted by T in the EVAL column of Table 1). This corresponds to the classical $k$-Means setting. In practice, however, one is often also interested in the *generalization error*. For the other half of the data sets, we retain 250,000 data points as the holdout set for the evaluation (denoted by H in the EVAL column of Table 1).

For all methods, we record the solution quality (either on the full data set or the holdout set) and measure the number of distance evaluations needed to run the algorithm. For ASSUMPTION-FREE K-MC$^2$ this includes both the preprocessing and the main loop. We run every algorithm 200 times with different random seeds and average the results. We further compute and display $95\%$ confidence intervals for the solution quality.

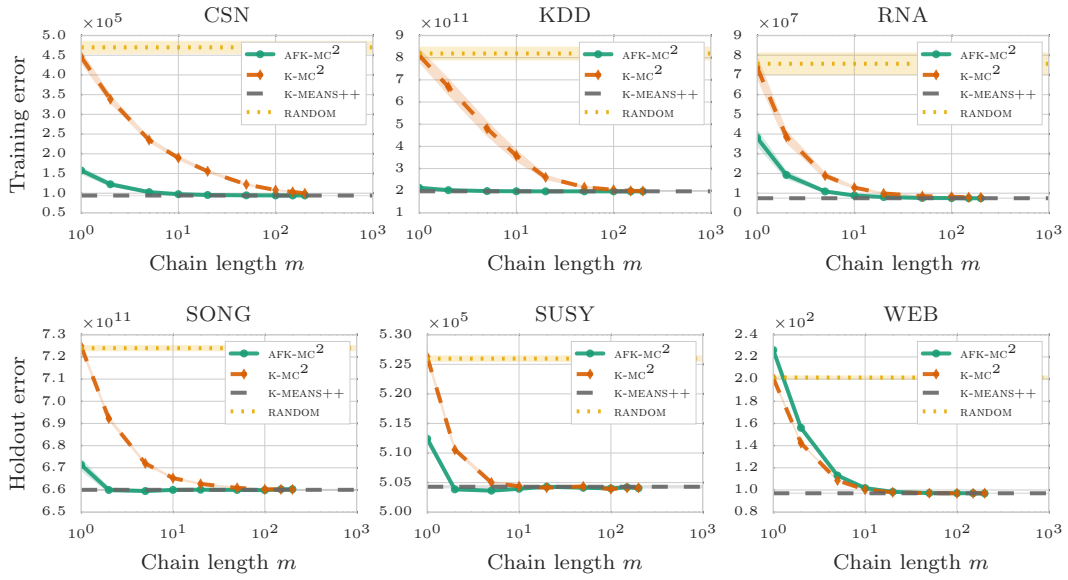

Figure 1: Quantization error in relation to the chain length $m$ for ASSUMPTION-FREE K-MC$^2$ and K-MC$^2$ as well as the quantization error for k-means++ and RANDOM (with no dependence on $m$). ASSUMPTION-FREE K-MC$^2$ substantially outperforms K-MC$^2$ except on WEB. Results are averaged across 200 iterations and shaded areas denote 95% confidence intervals.

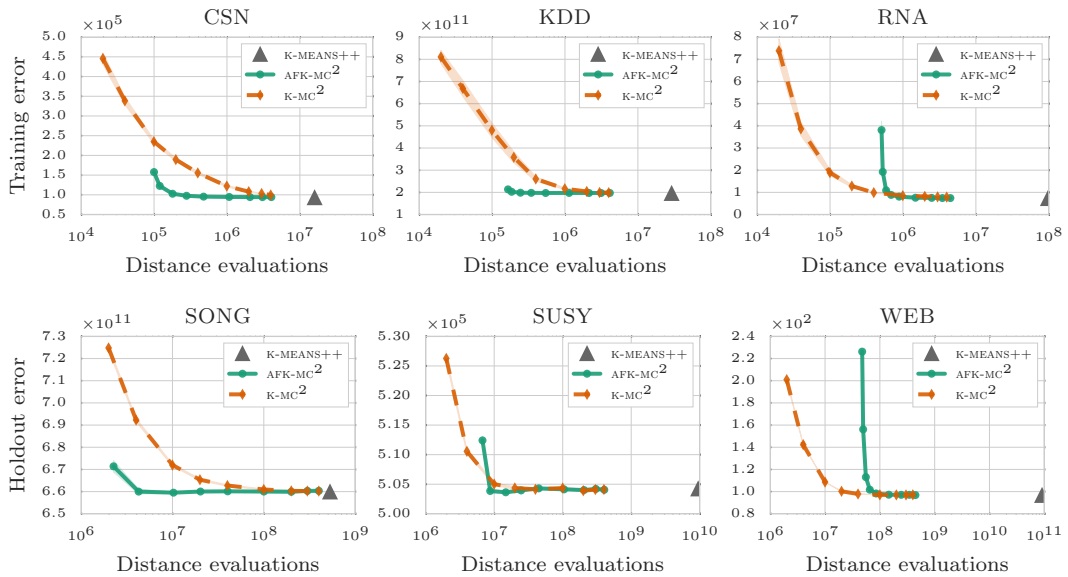

Figure 2: Quantization error in relation to the number of distance evaluations for ASSUMPTION-FREE K-MC$^2$, K-MC$^2$ and k-means++. ASSUMPTION-FREE K-MC$^2$ provides a speedup of up to several orders of magnitude compared to k-means++. Results are averaged across 200 iterations and shaded areas denote 95% confidence intervals.

Table 3: Relative speedup (in terms of distance evaluations) in relation to `k-means++`.

|  | CSN | KDD | RNA | SONG | SUSY | WEB |
|---|---|---|---|---|---|---|
| K-MEANS++ | 1.0× | 1.0× | 1.0× | 1.0× | 1.0× | 1.0× |
| K-MC$^2$ ($m = 20$) | 40.0× | 72.9× | 244.3× | 13.3× | 237.5× | 2278.1× |
| K-MC$^2$ ($m = 100$) | 8.0× | 14.6× | 48.9× | 2.7× | 47.5× | 455.6× |
| K-MC$^2$ ($m = 200$) | 4.0× | 7.3× | 24.4× | 1.3× | 23.8× | 227.8× |
| AFK-MC$^2$ ($m = 20$) | 33.3× | 53.3× | 109.7× | 13.2× | 212.3× | 1064.7× |
| AFK-MC$^2$ ($m = 100$) | 7.7× | 13.6× | 39.2× | 2.6× | 46.4× | 371.0× |
| AFK-MC$^2$ ($m = 200$) | 3.9× | 7.0× | 21.8× | 1.3× | 23.5× | 204.5× |

**Discussion.** Figure 1 shows the expected quantization error for the two baselines, RANDOM and `k-means++`, and for the MCMC methods with different chain lengths $m$. As expected, the seeding step of `k-means++` strongly outperforms RANDOM on all data sets. As the chain length $m$ increases, the quality of solutions produced by both ASSUMPTION-FREE K-MC$^2$ and K-MC$^2$ quickly converges to that of `k-means++` seeding.

On all data sets except WEB, ASSUMPTION-FREE K-MC$^2$ starts with a lower initial error due to the improved proposal distribution and outperforms K-MC$^2$ for any given chain length $m$. For WEB, both algorithms exhibit approximately the same performance. This is expected as $\alpha(\mathcal{X})$ of WEB is very low (see Table 1). Hence, there is only a minor difference between the nonuniform proposal of ASSUMPTION-FREE K-MC$^2$ and the uniform proposal of K-MC$^2$. In fact, one of the key advantages of ASSUMPTION-FREE K-MC$^2$ is that its proposal adapts to the data set at hand.

As discussed in Section 3, ASSUMPTION-FREE K-MC$^2$ requires an additional preprocessing step to compute the nonuniform proposal. Figure 2 shows the expected solution quality in relation to the total computational complexity in terms of number of distance evaluations. Both K-MC$^2$ and ASSUMPTION-FREE K-MC$^2$ generate solutions that are competitive with those produced by the seeding step of `k-means++`. At the same time, they do this at a fraction of the computational cost. Despite the preprocessing, ASSUMPTION-FREE K-MC$^2$ clearly outperforms K-MC$^2$ on the data sets with large values for $\alpha(\mathcal{X})$ (CSN, KDD and SONG). The additional effort of computing the nonuniform proposal is compensated by a substantially lower expected quantization error for a given chain size. For the other data sets, ASSUMPTION-FREE K-MC$^2$ is initially disadvantaged by the cost of computing the proposal distribution. However, as $m$ increases and more time is spent computing the Markov chains, it either outperforms K-MC$^2$ (RNA and SUSY) or matches its performance (WEB).

Table 3 details the practical significance of the proposed algorithm. The results indicate that in practice it is sufficient to run ASSUMPTION-FREE K-MC$^2$ with a chain length independent of $n$. Even with a small chain length, ASSUMPTION-FREE K-MC$^2$ produces competitive clusterings at a fraction of the computational cost of the seeding step of `k-means++`. For example on CSN, ASSUMPTION-FREE K-MC$^2$ with $m = 20$ achieves a relative error of $1.45\%$ and a speedup of $33.3\times$. At the same time, K-MC$^2$ would have exhibited a substantial relative error of $65.34\%$ while only obtaining a slightly better speedup of $40.0\times$.

## 5 Conclusion

In this paper, we propose ASSUMPTION-FREE K-MC$^2$, a simple and fast seeding algorithm for $k$-Means. In contrast to the previously introduced algorithm K-MC$^2$, it produces provably good clusterings even without assumptions on the data. As a key advantage, ASSUMPTION-FREE K-MC$^2$ allows to provably trade off solution quality for a decreased computational effort. Extensive experiments illustrate the practical significance of the proposed algorithm: It obtains competitive clusterings at a fraction of the cost of `k-means++` seeding and it outperforms or matches its main competitor K-MC$^2$ on all considered data sets.

**Acknowledgments**

This research was partially supported by ERC StG 307036, a Google Ph.D. Fellowship and an IBM Ph.D. Fellowship.

## Footnotes

[1] Let $\Omega$ be a finite sample space on which two probability distributions $p$ and $q$ are defined. The total variation distance $\|p - q\|_{\text{TV}}$ between $p$ and $q$ is given by $\frac{1}{2} \sum_{x \in \Omega} |p(x) - q(x)|$.

[2]An implementation of ASSUMPTION-FREE K-MC$^2$ has been released at `http://olivierbachem.ch`.

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
