[Supplementary Material]

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

++ | $1.0\times$ | $1.0\times$ | $1.0\times$ | $1.0\times$ | $1.0\times$ | $1.0\times$ |
| K-MC$^2$ ($m = 20$) | $40.0\times$ | $72.9\times$ | $244.3\times$ | $13.3\times$ | $237.5\times$ | $2278.1\times$ |
| K-MC$^2$ ($m = 100$) | $8.0\times$ | $14.6\times$ | $48.9\times$ | $2.7\times$ | $47.5\times$ | $455.6\times$ |
| K-MC$^2$ ($m = 200$) | $4.0\times$ | $7.3\times$ | $24.4\times$ | $1.3\times$ | $23.8\times$ | $227.8\times$ |
| AFK-MC$^2$ ($m = 20$) | $33.3\times$ | $53.3\times$ | $109.7\times$ | $13.2\times$ | $212.3\times$ | $1064.7\times$ |
| AFK-MC$^2$ ($m = 100$) | $7.7\times$ | $13.6\times$ | $39.2\times$ | $2.6\times$ | $46.4\times$ | $371.0\times$ |
| AFK-MC$^2$ ($m = 200$) | $3.9\times$ | $7.0\times$ | $21.8\times$ | $1.3\times$ | $23.5\times$ | $204.5\times$ |

**Discussion.** Figure 1 shows the expected quantization error for the two baselines, RANDOM and `k-means++`, and for the MCMC methods with different chain lengths $m$. As expected, the seeding step of `k-means++` strongly outperforms RANDOM on all data sets. As the chain length $m$ increases, the quality of solutions produced by both ASSUMPTION-FREE K-MC$^2$ and K-MC$^2$ quickly converges to that of `k-means++` seeding.

On all data sets except WEB, ASSUMPTION-FREE K-MC$^2$ starts with a lower initial error due to the improved proposal distribution and outperforms K-MC$^2$ for any given chain length $m$. For WEB, both algorithms exhibit approximately the same performance. This is expected as $\alpha(\mathcal{X})$ of WEB is very low (see Table 1). Hence, there is only a minor difference between the nonuniform proposal of ASSUMPTION-FREE K-MC$^2$ and the uniform proposal of K-MC$^2$. In fact, one of the key advantages of ASSUMPTION-FREE K-MC$^2$ is that its proposal adapts to the data set at hand.

As discussed in Section 3, ASSUMPTION-FREE K-MC$^2$ requires an additional preprocessing step to compute the nonuniform proposal. Figure 2 shows the expected solution quality in relation to the total computational complexity in terms of number of distance evaluations. Both K-MC$^2$ and ASSUMPTION-FREE K-MC$^2$ generate solutions that are competitive with those produced by the seeding step of `k-means++`. At the same time, they do this at a fraction of the computational cost. Despite the preprocessing, ASSUMPTION-FREE K-MC$^2$ clearly outperforms K-MC$^2$ on the data sets with large values for $\alpha(\mathcal{X})$ (CSN, KDD and SONG). The additional effort of computing the nonuniform proposal is compensated by a substantially lower expected quantization error for a given chain size. For the other data sets, ASSUMPTION-FREE K-MC$^2$ is initially disadvantaged by the cost of computing the proposal distribution. However, as $m$ increases and more time is spent computing the Markov chains, it either outperforms K-MC$^2$ (RNA and SUSY) or matches its performance (WEB).

Table 3 details the practical significance of the proposed algorithm. The results indicate that in practice it is sufficient to run ASSUMPTION-FREE K-MC$^2$ with a chain length independent of $n$. Even with a small chain length, ASSUMPTION-FREE K-MC$^2$ produces competitive clusterings at a fraction of the computational cost of the seeding step of `k-means++`. For example on CSN, ASSUMPTION-FREE K-MC$^2$ with $m = 20$ achieves a relative error of $1.45\%$ and a speedup of $33.3\times$. At the same time, K-MC$^2$ would have exhibited a substantial relative error of $65.34\%$ while only obtaining a slightly better speedup of $40.0\times$.

## 5 Conclusion

In this paper, we propose ASSUMPTION-FREE K-MC$^2$, a simple and fast seeding algorithm for $k$-Means. In contrast to the previously introduced algorithm K-MC$^2$, it produces provably good clusterings even without assumptions on the data. As a key advantage, ASSUMPTION-FREE K-MC$^2$ allows to provably trade off solution quality for a decreased computational effort. Extensive experiments illustrate the practical significance of the proposed algorithm: It obtains competitive clusterings at a fraction of the cost of `k-means++` seeding and it outperforms or matches its main competitor K-MC$^2$ on all considered data sets.

**Acknowledgments**

This research was partially supported by ERC StG 307036, a Google Ph.D. Fellowship and an IBM Ph.D. Fellowship.

## Footnotes

[1]Let $\Omega$ be a finite sample space on which two probability distributions $p$ and $q$ are defined. The total variation distance $\|p - q\|_{\mathrm{TV}}$ between $p$ and $q$ is given by $\frac{1}{2}\sum_{x \in \Omega} |p(x) - q(x)|$.

[2]An implementation of ASSUMPTION-FREE K-MC$^2$ has been released at `http://olivierbachem.ch`.

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

# A    Proof of Theorem 1

We first derive two auxiliary lemmas and then use them to derive the main claim in Theorem 1. Intuitively, we first bound the difference between exact $D^2$-sampling and a single Markov chain in terms of the total variation distance. Then, we analyze how the approximation error accumulates across iterations and bound the expected solution quality.

**Single Markov chain.** Algorithm 1 samples an initial center $c_1 \in \mathcal{X}$ uniformly at random and then uses it to derive the proposal distribution $q(x \mid c_1)$ as defined in (4). In each iteration $i = 2, 3, \ldots, k$, the Markov chain of length $m$ implicitly defines a probability distribution over all $x \in \mathcal{X}$. Given any set of previously sampled centers $C \subseteq \mathcal{X}$ with $c_1 \in C$, we denote by $\tilde{p}_m^{c_1}(x \mid C)$ the implied probability of sampling a point $x \in \mathcal{X}$ using this Markov chain. Correspondingly, $p(x \mid C)$ as defined in (1) denotes the probability of adding a point $x \in \mathcal{X}$ to the set $C \subseteq \mathcal{X}$ when using $D^2$-sampling. The following result shows that in any iteration either $C$ is $\epsilon_1$-competitive compared to $c_1$ or the Markov chain approximates $D^2$-sampling well in terms of total variation distance.

**Lemma 1.** *Let $\epsilon_1, \epsilon_2 \in (0, 1)$ and $c_1 \in \mathcal{X}$. Consider any set $C \subseteq \mathcal{X}$ with $c_1 \in C$. For $m \geq 1 + \frac{2}{\epsilon_1} \log \frac{1}{\epsilon_2}$, at least one of the following must hold:*

- *(i) $\phi_C(\mathcal{X}) < \epsilon_1 \phi_{c_1}(\mathcal{X})$, or*

- *(ii) $\|p(\cdot \mid C) - \tilde{p}_m^{c_1}(\cdot \mid C)\|_{\mathrm{TV}} \leq \epsilon_2$.*

*Proof of Lemma 1.*  Consider fixed $c_1$ and $C$. By design, the stationary distribution of the Markov chain is $p(\cdot \mid C)$. Using Corollary 1 of Cai (2000), convergence in terms of total variation distance is bounded by

$$\|p(\cdot \mid C) - \tilde{p}_m^{c_1}(\cdot \mid C)\|_{\mathrm{TV}} \leq \left(1 - \frac{1}{\gamma}\right)^{m-1} \leq e^{-\frac{m-1}{\gamma}}$$

where

$$\gamma = \max_{x \in \mathcal{X}} \frac{p(x \mid C)}{q(x \mid c_1)}.$$

For a chain length $m \geq 1 + \gamma \log \frac{1}{\epsilon_2}$, the total variation is hence bounded by $\epsilon_2$.

For $\phi_C(\mathcal{X}) \geq \epsilon_1 \phi_{c_1}(\mathcal{X})$, we have using (1) and (4) that

$$\frac{p(x \mid C)}{q(x \mid c_1)} \leq 2 \frac{\mathrm{d}(x, C)^2}{\mathrm{d}(x, c_1)^2} \frac{\phi_{c_1}(\mathcal{X})}{\phi_C(\mathcal{X})} \leq \frac{2}{\epsilon_1}, \quad \forall x \in \mathcal{X}$$

since $\mathrm{d}(x, C)^2 \leq \mathrm{d}(x, c_1)^2$ and thus $\gamma \leq 2/\epsilon_1$. Therefore either (i) or (ii) in Lemma 1 has to hold since the choice of $c_1$ and $C$ was arbitrary and $c_1 \in C$. $\square$

**Expected quantization error across iterations.**  Again, consider a fixed initial center $c_1$ and a set of previously sampled centers $C \subseteq \mathcal{X}$ that includes $c_1$. We denote by $A^{c_1}(C, \ell)$ the expected quantization error of sequentially adding $\ell$ points to $C$ using the Markov chains defined in lines 6 - 11 of Algorithm 1. For $\ell \in \mathbb{N}$ and $C \subseteq \mathcal{X}$ with $c_1 \in C$, we have

$$A^{c_1}(C, \ell) = \sum_{x \in \mathcal{X}} \tilde{p}_m^{c_1}(x \mid C) A^{c_1}(C \cup \{x\}, \ell - 1) \tag{5}$$

as well as $A^{c_1}(C, 0) = \phi_C(\mathcal{X})$.

For $\ell \in \mathbb{N}$ and $C \subseteq \mathcal{X}$ with $c_1 \in C$, we further define $P^{c_1}(C, \ell)$ by

$$P^{c_1}(C, 0) = \mathbf{1}_{\{\phi_C(\mathcal{X}) < \epsilon_1 \phi_{c_1}(\mathcal{X})\}}, \quad \text{and}$$

$$P^{c_1}(C, \ell) = \sum_{x \in \mathcal{X}} \tilde{p}_m^{c_1}(x \mid C) P^{c_1}(C \cup \{x\}, \ell - 1).$$

Intuitively, $P^{c_1}(C, \ell)$ captures the probability of sampling a solution that is $\epsilon_1$-competitive with $\phi_{c_1}(\mathcal{X})$ when adding $\ell$ centers to $C$ using the Markov chains.

Finally, for $\ell \in \mathbb{N}_0$ and a set $C \subseteq \mathcal{X}$, let $B(C, \ell)$ denote the expected quantization error of sequentially adding $\ell$ points to $C$ using $D^2$-sampling. For $\ell \in \mathbb{N}$ and $C \subseteq \mathcal{X}$, we have

$$B(C, \ell) = \sum_{x \in \mathcal{X}} p(x \mid C) B(C \cup \{x\}, \ell - 1). \tag{6}$$

as well as $B(C, 0) = A^{c_1}(C, 0) = \phi_C(\mathcal{X}), \forall c_1 \in \mathcal{X}$.

The following result bounds $A^{c_1}(C, \ell)$ in terms of $B(C, \ell)$, $P^{c_1}(C, \ell)$ and $\phi_{c_1}(\mathcal{X})$.

**Lemma 2.** *Let $\epsilon_1, \epsilon_2 \in (0, 1)$ and $c_1 \in \mathcal{X}$. Consider any set $C \subseteq \mathcal{X}$ that satisfies $c_1 \in C$. For $\ell \in \mathbb{N}_0$ and $m \geq 1 + \frac{2}{\epsilon_1} \log \frac{1}{\epsilon_2}$, it then holds that*

$$A^{c_1}(C, \ell) \leq \left(\epsilon_1 P^{c_1}(C, \ell) + \ell \epsilon_2\right) \phi_{c_1}(\mathcal{X}) + B(C, \ell).$$

*Proof of Lemma 2.* For $\ell = 0$, the claim holds trivially since $A^{c_1}(C, \ell) = B(C, \ell)$. For $\ell \in \mathbb{N}$ we prove the result by induction, i.e., we assume that it holds for $\ell - 1$ and show that it holds for $\ell$.

Consider any initial center $c_1$ and any set of centers $C \subseteq \mathcal{X}$ with $c_1 \in C$. If we have $\phi_C(\mathcal{X}) < \epsilon_1 \phi_{c_1}(\mathcal{X})$, the claim holds again trivially since $P^{c_1}(C, \ell) = 1$ and $A^{c_1}(C, \ell) \leq \phi_C(\mathcal{X})$. For the remainder of the proof, we thus assume $\phi_C(\mathcal{X}) \geq \epsilon_1 \phi_{c_1}(\mathcal{X})$.

We define the two sets

$$\mathcal{X}_A = \left\{x \in \mathcal{X} \mid \phi_{C \cup \{x\}}(\mathcal{X}) < \epsilon_1 \phi_{c_1}(\mathcal{X})\right\}$$
$$\mathcal{X}_B = \left\{x \in \mathcal{X} \mid \phi_{C \cup \{x\}}(\mathcal{X}) \geq \epsilon_1 \phi_{c_1}(\mathcal{X})\right\}$$

and note that $\mathcal{X}_B = \mathcal{X} \setminus \mathcal{X}_A$. Intuitively, $\mathcal{X}_A$ denotes the part of $\mathcal{X}$ that, if sampled, leads to $\epsilon_1$-competitive solutions with regards to $c_1$. This allows us to split (5) into

$$A^{c_1}(C, \ell) = \underbrace{\sum_{x \in \mathcal{X}_A} \tilde{p}_m^{c_1}(x \mid C) A^{c_1}(C \cup \{x\}, \ell - 1)}_{(*)} + \underbrace{\sum_{x \in \mathcal{X}_B} \tilde{p}_m^{c_1}(x \mid C) A^{c_1}(C \cup \{x\}, \ell - 1)}_{(**)}.$$

Setting $\pi = \sum_{x \in \mathcal{X}_A} \tilde{p}_m^{c_1}(x \mid C)$ we obtain

$$(*) = \sum_{x \in \mathcal{X}_A} \tilde{p}_m^{c_1}(x \mid C) A^{c_1}(C \cup \{x\}, \ell - 1) \leq \pi \epsilon_1 \phi_{c_1}(\mathcal{X}). \tag{7}$$

By the definitions of $P^{c_1}(C, \ell)$ and $\pi$ we have

$$\sum_{x \in \mathcal{X}_B} \tilde{p}_m^{c_1}(x \mid C) P^{c_1}(C \cup \{x\}, \ell - 1) = P^{c_1}(C, \ell) - \pi.$$

To bound the term $(**)$, we now assume that Lemma 2 holds for $\ell - 1$. Using the previous statement and the fact that $1 - \pi \leq 1$, we obtain

$$(**) \leq \left[(P^{c_1}(C, \ell) - \pi)\epsilon_1 + (\ell - 1)\epsilon_2\right] \phi_{c_1}(\mathcal{X}) + \underbrace{\sum_{x \in \mathcal{X}_B} \tilde{p}_m^{c_1}(x \mid C) B(\{x\} \cup C, \ell - 1)}_{(***)}. \tag{8}$$

Since $B(\{x\} \cup C, \ell - 1) \geq 0$ and $\mathcal{X}_B \subseteq \mathcal{X}$, we have

$$(***) \leq \sum_{x \in \mathcal{X}} \tilde{p}_m^{c_1}(x \mid C) B(\{x\} \cup C, \ell - 1).$$

Since $\phi_{(\cdot)}(\mathcal{X})$ is monotone, adding any center $x \in \mathcal{X}$ to $C$ can only reduce the quantization error and hence $B(\{x\} \cup C, \ell - 1) \leq \phi_{c_1}(\mathcal{X})$. Furthermore, for all $x \in \mathcal{X}$

$$\tilde{p}_m^{c_1}(x \mid C) \leq p(x \mid C) + [\tilde{p}_m^{c_1}(x \mid C) - p(x \mid C)]^+.$$

This implies

$$(***) \leq \sum_{x \in \mathcal{X}} p(x \mid C) B(\{x\} \cup C, \ell - 1) + \sum_{x \in \mathcal{X}} [\tilde{p}_m^{c_1}(x \mid C) - p(x \mid C)]^+ \phi_{c_1}(\mathcal{X}).$$

Using (6) and Lemma 1, it holds that

$$(* * *) \leq B(C, \ell) + \underbrace{\|\tilde{p}_m^{c_1}(x \mid C) - p(x \mid C)\|_{\mathrm{TV}}}_{\leq \epsilon_2 \text{ by Lemma 1}} \phi_{c_1}(\mathcal{X}).$$

Combining this with (7) and (8) proves the Lemma, i.e.,

$$A^{c_1}(C, \ell) \leq (\epsilon_1 P^{c_1}(C, \ell) + \ell \epsilon_2) \phi_{c_1}(\mathcal{X}) + B(C, \ell).$$

$\square$

With this result we are now able to prove the main claim in Theorem 1.

*Proof of Theorem 1.* Using $P^{c_1}(c_1, k - 1) \leq 1$ and Lemma 2 with $\epsilon_1 = \epsilon/4$ and $\epsilon_2 = \epsilon/4k$, we have

$$A^{c_1}(c_1, k - 1) \leq (\epsilon_1 P^{c_1}(C, k - 1) + (k - 1)\epsilon_2) \phi_{c_1}(\mathcal{X}) + B(C, k - 1) \leq \frac{\epsilon}{2} \phi_{c_1}(\mathcal{X}) + B(C, k - 1)$$

By Theorem 1.1 of Arthur & Vassilvitskii (2007), we have

$$\frac{1}{|\mathcal{X}|} \sum_{c_1 \in \mathcal{X}} B(\{c_1\}, k - 1) \leq 8 (\log_2 k + 2) \phi_{OPT}^k(\mathcal{X}).$$

Furthermore, by Lemma 3.1 of Arthur & Vassilvitskii (2007)

$$\frac{1}{|\mathcal{X}|} \sum_{c_1 \in \mathcal{X}} \phi_{c_1}(\mathcal{X}) = 2 \operatorname{Var}(\mathcal{X}).$$

This allows us to obtain the required solution quality, i.e.,

$$\mathbb{E}\left[\phi_{\mathrm{AFK\text{-}MC}^2}\right] = \frac{1}{|\mathcal{X}|} \sum_{c_1 \in \mathcal{X}} A^{c_1}(\{c_1\}, l) \leq 8 (\log_2 k + 2) \phi_{OPT}^k(\mathcal{X}) + \epsilon \operatorname{Var}(\mathcal{X}).$$

The computational complexity of the preprocessing step is a single pass through the data, i.e., $nd$ where $n$ is the number of data points and $d$ its dimensionality. For each iteration $i = 2, 3, \ldots, k$, the computational complexity of constructing the Markov chain is $\mathcal{O}(im)$. Hence, the computational complexity of the main loop is $\mathcal{O}\left(\frac{1}{\epsilon} k^2 d \log \frac{k}{\epsilon}\right)$. $\square$