[Reviews · NeurIPS 2016]

Reviewer 1

Summary

This paper extends the work of Bachem et. al. in reducing the running time of the k-means++ procedure. Bachem et. al. suggested sampling points using a Markov process (instead of computing the complete distribution after every iteration and then sampling from this distribution). Since computing the distribution is a costly operation, this technique reduces the running time significantly for very large datasets. However, in order to show that the sampling probabilities is similar to that of the k-means++ procedure, we need to show bounds on the mixing time of the Markov process. This in turn imposes constraints on the datasets on which this technique may be applied. The current paper approaches the problem in a slightly different manner. They try to argue that the Markov process based algorithm works for any dataset if one allows some additive factors in the approximation guarantee. They also argue that in some cases this additive approximation factor does not cause any serious problems. Furthermore, they show that on many real datasets, the additive term is small for reasonable values of parameters. As in Bachem et. al.’s work, the process is significantly faster than the k-means++ seeding procedure.

Qualitative Assessment

This paper is well written and may be followed and appreciated by non-experts. In my opinion, the results in the paper adds to our knowledge about the k-means++ procedure.

Confidence in this Review

2-Confident (read it all; understood it all reasonably well)


Reviewer 2

Summary

Authors identify a superior proposal distribution for an MCMC-based seeding strategy for k-means. Surprisingly (!), only a single data pass of preprocessing based upon a randomly chosen center is sufficient to define a proposal distribution which has fast convergence for all iterations.

Qualitative Assessment

The simplicity and efficacy of this approach suggests it is likely to be the method of choice in the single-node setting. If a kmeans|| style oversampling trick can be applied to loop on line 5 of algorithm 1, then it would be a strong contender in the distributed setting as well. It is unclear in the experimental section if relative errors are reported for just the seeding step, or if Lloyd's algorithm has been applied to refine the solutions. Assuming the former, it is unclear the extent to which the differences in table 2 would persist given even one Lloyd iteration (except, perhaps, wrt random initialization).

Confidence in this Review

2-Confident (read it all; understood it all reasonably well)


Reviewer 3

Summary

The initialization, or seeding of a K-means algorithm, when k-initial cluster centers are assigned, is a crucial step - as the accuracy of the clustering varies greatly with this. The D^2-sampling for seeding is an well-known procedure that provably outperform random clustering. However this algorithm does not scale with the size of the dataset, which is especially disadvantageous for massive datasets (complexity ~ nkd, n data points, k clusters, d dimension). This paper provides an alternative sampling method that has complexity ~ mk^2d (m is a system parameter). The algorithm is a modification of an existing method called k_MC^2, which has the same complexity, but has theoretical guarantee only assuming an random independent model for dataset. While the practical implication for this result may be limited, it is a crucial theoretical step. However this modified algorithm perform even better experimentally.

Qualitative Assessment

The proof of the main result, theorem 1 seems quite straightforward. There seem not to be new technical challenges in the proof (or it is not explicitly mentioned). In theorem 1, the error guarantee contains a variance term for the dataset - I assume that is the empirical variance. However this term is not there in the D^2 sampling guarantee. This means there may be dataset where this algorithm may be quite bad. Now, obviously such datasets can be avoided during experiment set up. It is not clear to me how bad the implication of this term can be.

Confidence in this Review

2-Confident (read it all; understood it all reasonably well)


Reviewer 4

Summary

The paper proposed a MCMC approach to approximate the D^2-sampling (k-means++) for finding k centers in clustering, by modifying the proposal distribution of an earlier paper by Banchem 16. The algorithm is able to approximately maintain the O(log k) approximation guarantee to the k-means objective for any dataset (as is k-means++), if their analysis is correct. The running time of the algorithm is linear in the number of data size, but shaves off a factor of k (the number of clusters) comparing to that of k-means++.

Qualitative Assessment

Technical quality: I have two doubts regarding the proof of their Lemma 2: 1. The authors argue, in the case of \phi_{C}(X) < \epsilon_1\phi_{c_1}, the claim holds trivially. I don't see how this goes through. First, I don't see why A^{c_1}(C,l) < =\phi_{C}(X). As I understand it, A^{c_1}(C,l) is the expected cost of C while \phi_{C}(X) is the actual cost of C (random quantity). If so, why is the former upper bounded by the latter? Second, even if the relation above holds, I don't see how the statement holds, since then we'll get A^{c_1}(C,l) < =\phi_{C}(X) < \epsilon_1\phi_{c_1}(X). How does this imply the statement using P^{c_1}(C,l) < 1? 2. I don't see how inequality (8) holds, since I can't see why the equation right above it holds, using the definition of \pi. Novelty: I think the idea of using MCMC methods to approximate D^2-sampling scheme is great. But since this paper is not the first to propose this approach, and the theoretical analysis seems incremental to that of Banchem 16, I think I should take some points off here. Potential impact or usefulness: I think this paper could potentially have good impact in large-scale clustering applications, given that k-means++ is widely used together with Lloyd's algorithm for clustering. Although the proposed algorithm still has a linear running time dependence on the data size, I think the fact that they can shave off a factor of k while approximately maintaining the performance guarantee of k-means++ could mean a lot in practice. The proposed algorithm also seems simple enough to be implemented in practice, comparing to other scaled versions of k-means++. In general, I think this research direction worths further exploration. Clarity and presentation: In terms of highlighting their contribution to previous work, the paper does a good job via both writing and experiments. In terms of the analysis, I think the authors should make more efforts in clarifying definitions and providing For example, the definitions of \phi_{C}(X), A^{c^1}(C,l) and P^{c^1}(C,l) are unclear to me, and seem to be inconsistent from one proof to another. Partly because the definition of C at different stages of the algorithm can change. A^{c^1}(C,l) and P^{c^1}(C,l) became more unclear to me when the authors refer the readers to lines 6-11 of the algorithm; for a moment, I thought "l" denotes the length of the markov chain at an iteration. Maybe the authors can remove this reference. Also, I'd be great if the authors can provide more intuition of their analysis, on a high level.

Confidence in this Review

2-Confident (read it all; understood it all reasonably well)


Reviewer 5

Summary

The authors propose scalable to large datasets algorithm for finding initial cluster centers for k-means. The algorithm does not make a-priory assumptions about the data and its performance is demonstrated on several datasets.

Qualitative Assessment

The paper is well written and provides both the theoretical proof, and evaluation of the performance using several datasets. I find the approach useful. In most cases, the performance is superior to that of K-MC^2

Confidence in this Review

1-Less confident (might not have understood significant parts)


Reviewer 6

Summary

An algorithm for the seeding step of k-Means is proposed for the case of massive datasets.It is mainly based on a previous work (Bachem et al. 2016), which constructs a Markov chain to sample centers. The main novelty is the definition of the distribution used to constuct the Markov chain. The quality of solutions is bounded ; this constitutes the statement of a theorem. A study is carried out with real data sets. The results are compared with those obtained with three other seeding strategies. Competitive clusterings are obtained and the computational cost can be considerably reduced, or similar according to the considered data sets and seeding strategies.

Qualitative Assessment

The paper is quite interesting and clear. About experimental results, it would be necessary to describe the data sets, in order to explain what is the meaning of the different clusters. On line 236, "distance evaluations" should be replaced by "number of distance evaluations". In the computational complexity, the time required to sample a point using a distribution is not taken in consideration, whereas this time is certainly not negligible. I think the real computational time would be a better measure to compare the different seeding techniques.

Confidence in this Review

2-Confident (read it all; understood it all reasonably well)